# Effectiveness of *SaBang-DolGi* Walking Exercise Program on Physical and Mental Health of Menopausal Women

**DOI:** 10.3390/ijerph17186935

**Published:** 2020-09-22

**Authors:** EunHee Noh, JiYoun Kim, MunHee Kim, EunSurk Yi

**Affiliations:** 1Department of Exercise Rehabilitation & Welfare, Gachon University, 191 Hombakmoero, Yeonsu-gu, Incheon 406-799, Korea; sneh7078@gc.gachon.ac.kr (E.N.); yies@gachon.ac.kr (E.Y.); 2Department of Health Science, Korea National Sport University, Seoul 05541, Korea; kmoonhee@hanmail.net

**Keywords:** *SaBang-DolGi* walking, walking exercise, menopause, women’s health, physical, mental, health programs

## Abstract

Objective: We investigated the effectiveness of a 12-week *SaBang-DolGi* walking exercise program on the physical and mental health of menopausal women and aimed to provide the basic data needed to develop health promotion programs for the active and healthy aging of menopausal women. Materials and methods: The participants comprised 40 women aged 50–65 years who were divided into two randomly selected groups in training sessions (exercising group, *n* = 21 and control group, *n* = 19). A physical (grip, muscle and endurance) test and mental health test (simple mental health test II) were conducted using questionnaires with the aim of examining subjects’ physical and mental health before and after exercise. Results: After the intervention, the participants experienced positive changes in the physical dimension, with significant enhancements particularly in mental well-being and menopause-related health and subdomains. Controlled and regular exercise for 12 weeks was significantly correlated with a positive change in vitality and mental health. Conclusions: We found that the *SaBang-DolGi* walking exercise program helps to promote the physical and mental health of menopausal women who are exposed to the various stresses and depressions that accompany physical deterioration; the program was found to encourage active and healthy aging.

## 1. Introduction

Aging is an inevitable process in humans—even though there are individual differences—and menopause, in particular, is an important event for all women in relation to aging. Although the menopausal transition is a natural change, the accompanying symptoms and diseases before and after menopause can affect the maintenance of daily life and health [1]. During menopause, estrogen deficiency becomes noticeable because of the deterioration of ovarian function, leading to physical symptoms such as vasomotor symptoms including facial flushing, sweating, sleep disorders, joint and muscle disorders, changes in weight and fat distribution and physical symptoms, including reduced physical strength. In addition, menopause is accompanied by mental problems such as depression, instability, loss of motivation, hypothymia, memory loss and mental fatigue [2]. This is understood as an increased risk of health problems due to menopause, and 50–80% of women experience at least one symptom, lasting from about five years before menopause to seven years after menopause [3,4]. In short, the negative effects of the physical and mental symptoms during menopause may interfere with women’s ability to cope with the menopausal transition, which may cause women to have difficulties in continuing their healthy life [5]. To alleviate these symptoms, alternative therapies are widely used and show beneficial effects [6,7], and exercise therapy is being experimented with as an intervention method because it does not have side effects, it is inexpensive and it helps not only physical health but also mental health [8,9].Exercise has been proven to be effective in reducing menopausal symptoms and increasing muscle strength [10,11,12,13], reducing stress during menopause [14], reducing tension and anger to decrease depression [15], reducing anxiety [16,17] and recovering from low self-esteem [18]. In prior studies, participation in specialized physical activity programs, especially regular walking and physical stimulation exercises, have been found to be effective for individuals’ physical strength, physical and mental health [19]. Physical activity during the menopausal transition and post-menopause period has been reported to improve mental health, prevent weight gain, increase bone mineral density and muscle mass and reduce and the risks of other diseases (e.g., cancer, diabetes and heart disease) [8,20]. As an alternative intervention to maintain the physical and mental health of menopausal women, the benefits of exercise for health have been well established [21], including in terms of their long-term effectiveness [22].

Nevertheless, women experiencing the menopause are often reluctant to engage in exercise because of a lack of adequate opportunities, lack of access to facilities, high cost, safety fears, busy schedules and difficulties in accessing the programs [23,24,25,26]; a sedentary occupation is often also maintained for the reason of a lack of motivation. Easy-to-access community exercise programs will bring health benefits that will help menopausal women to increase their activity and maintain their functions.

*SaBang-DolGi* is a traditional Korean exercise. In this tradition, participants repeat the movements, beginning while watching the front, by walking around in the same place in four directions—east, west, south and north—and maintaining their energy and center of gravity in the center of the body using rotation and directionality, thereby achieving emotional stability. The steps of *SaBang-DolGi* can be found in *SaBang-ChiGi*—the basic principle of Korean traditional dance—in which dancers move in all directions—right, left, up and down—beat by beat. The *SaBang-DolGi* motion, as an esthetic structure of Korean traditional dance, is performed step by step according to the principle of movement. The basic step is shown in Figure 1. In other words, humans responsive to the sky above and the Earth below reach a state of impartiality in which they can achieve harmony through mutual communion and interaction [27]. *SaBang-DolGi* has the advantage of increasing the brain’s momentum and cognitive activity because of the necessity of remembering the direction, the amount of rotation and the order of connection of movements; thus, it helps to increase memory and prevent dementia as well as improves muscle strength through repetitive movements without incurring boredom [28]. In addition, when the repetitive motion is rotated in all directions and the last position becomes the first starting position, the adaptation to the posture shown in the condition of the expected turn changes the motion and position from the center while reducing the error range of the stride, thusit can change the timing of the proper muscle activity, size and muscle reflex activity [29]. Rhythmic phrases and intensive repetitive movements have positive effects on bone mass, flexibility and mental health conditions such as depression, in a manner similar to lifestyle music therapy [30,31,32,33,34]. Repeated and intensive use of the full body, such as in *SaBang-DolGi*, has been reported to be important for the relocation of the motor field of the cerebral cortex [35]; furthermore, a reduction of depression has been reported due to the central nervous system inducing changes in the cerebral cortex level [36,37].

In this study, the *SaBang-DolGi* walking exercise was selected as an exercise program for menopausal women. The advantage is the ease of accessing the program, which means that it can be performed alone or in a group regardless of attire, place and time and without special tools. It is also considered to be a viable health program that can be safely performed in post-menopausal women who are exposed to the risk of obesity and joint injury because of a sedentary lifestyle, which is currently very common. Nevertheless, the only research based on the program is the study of women’s physical health factors, cardiovascular risk factors and quality of life after the menopause [38] and the static posture, walking variables and balancing ability of male college students [39].

This exercise was created in Korea based on the wellness life approach established in the early 2000s. The number of rehabilitation patients who felt better after this exercise has increased, and they spread the exercise through word of mouth. The exercise has the features of complex exercise, in which participants walk in the same place chanting slogans and create a harmony of predetermined motions at the same time with *SaBang-DolGi*. Currently, this practice is spreading as part of a project promoting physical activities and health, such as in welfare centers for the elderly, public health centers, medical insurance corporations and school sports programs by training professional instructors; however, there is a lack of scientific verification of the *SaBang-DolGi* walking exercise program.

Therefore, this study aimed to verify the effectiveness of the *SaBang-DolGi* walking exercise program on physical and mental health variables in menopausal women by providing them with a program which promotes active and healthy aging and can be performed even in limited spaces.

## 2. Materials and Methods

### 2.1. Participants

The present study was a randomized controlled trial. Forty eligible women were recruited and randomly assigned to either a 12-week exercise intervention (three sessions / week) or to a usual care (control) group. The sample size was determined using the G*Power program, considering a significance level of 0.05 and a power of 0.80 to obtain an effect size of 0.5. It was determined that a sample size of 34 individuals was adequate, meaning that our sample satisfied the conditions for the recommended sample size. The lack of serious illness and the consent to participate in the research were inclusion criteria; exclusion criteria comprised physical problems related to spinal cord injury, paralysis, history of antidepressant use, history of psychiatric disorders, history of hormonal therapy (HT) use and symptoms for all items of the MRS. Participants were fully informed of the purpose and content of the study. From 45 participants, five participants including those who received hormone replacement therapy or who had osteoporosis, disability and serious illness were excluded. In addition, only voluntary participants who signed a clinical trial consent form were allowed to participate, and an approval from the Gachon University’s Bioethics Committee (Approval No: 1044396-201909-HR-162-01) was obtained prior to the start of the study. Table 1 shows the physical characteristics of the participants in the study.

### 2.2. Study Design

In this study, participants were randomly assigned to two different groups. To analyze the effectiveness of the *SaBang-DolGi* walking exercise program, participants were divided into two groups: the experimental group—a group of participants who participated regularly for 3 h a week, for 60 min or more per session according to the exercise guidelines of the American College of Sports Medicine (ACSM) [40]—and a control group—a group of participants who did not participate in the exercise [40]. The study was designed, as shown in Figure 2.

### 2.3. Measurement

#### 2.3.1. Body Composition

The body compositions of the participants were measured between 9 AM and 1 PM after fasting, and heights and weights were measured using an automatic extensometer (BSM370, Inbody, Korea). The body mass index (BMI) was calculated using the formula Weight (kg)/Height (m^2^). The waist circumference was measured twice to 0.1 cm, from the midpoint of the iliac crest and the lower edge of the rib, when the participants stopped exhaling, and then the mean was calculated.

#### 2.3.2. Physical Health Test

The physical health test measured muscle strength, muscular endurance and flexibility. The grip strength was measured for muscle strength, a sit-up was performed for muscular endurance and a bend of the upper-body forward was performed to measure flexibility. The methods of measurement were as follows. For grip strength, a hand dynamometer (EH101; Camry, Zhongshan, China) was used. The subjects were instructed to hold the hand dynamometer with its needle facing outward, adjusting the handle to the second node of the finger, and participants then opened both legs and straightened their arms. They performed the task with the maximum force for 5 s while maintaining the posture with a signal from the operator. After the hand dynamometer was detached from the body, both hands were measured twice alternately, and a good record was recorded in 0.1-kg increments. For the muscular endurance measurement, participants were instructed to bend their knees, lying on the back with their feet about 30 cm away from their hips, cross their arms over their chests with their elbows and repeatedly touch their thigh with their elbows for 1 min and then the floor. Then, the number of motions was recorded. For the flexibility test, the subjects were instructed to take off their shoes and sit on a measuring instrument with both feet extended. They bent their upper body forward as much as possible without bending their knees. With both hands extended as far as possible, the numerical value was measured in a state in which they stopped for 2 s. The measurement was conducted twice, and the highest value was recorded in 0.1-cm increments. As a measuring tool, a sit-and-reach tester (New sit-and-reach tester DWR-OT1038; Dauri Sports, Seoul, Korea) was used.

#### 2.3.3. Mental Health Test

In this study, the simplified mental health test II (Korea Symptom-Checklist-90-Revision (SCL-95-R) was used to evaluate the mental health of participants. The simplified mental health test II (KSCL95) is a questionnaire that was made by revising and standardizing the study version of the “Symptom Checklist-90-Revision” (SCL-90-R; Derogatis, 1977) questionnaire developed by Dr. L.R. Derogatis [41], a psychologist at Johns Hopkins Medical School in the United States, in accordance with the current situation in Korea in 2015, under his approval. A four-point Likert scale (0: Not at all; 1: Occasionally, 2: Frequently, 3: Almost always) was used as a method of responding to the questionnaire, where a higher score meant more severe symptoms. Seventeen symptoms were evaluated in the domain of “emotional disorders” consisting of nine symptoms (depression, anxiety, panic attacks, agoraphobia, obsessive-compulsive, obsessive-compulsive personality trait, PTSD (Post-Traumatic Stress Disorder), aggression and somatization); the domain of “adaption to reality” consisted of three symptoms (manic episode, paranoia, schizophrenia); and the other domain consisted of five symptoms (suicide, addiction, sleep problems, stress vulnerability and low self-regulation). Since the test results were presented as T scores, the symptoms could be interpreted as follows: ≤T59, moderate (no-problem), T60–T69, caution required and risky; ≥T70, a problem (there are possible mental health problems).The measurement of the factor analysis for the survey items used the emotional area with a Cronbach’s α of 0.894, other areas with a Cronbach’s α of 0.863 and reality adaptation problems with a Cronbach’s α of 0.697; these values were mostly found to be 0.6 or higher, indicating reliability and validity (Table 2).

#### 2.3.4. SaBang-DolGi Walking Exercise Program

This program was conducted by a qualified professional instructor, and the instructor checked the motions and postures of the subjects. Exercise was performed in 3 sessions a week, for 60 min a day, for a total of 12 weeks. Before and after the exercise, warm-up and warm-down exercises were performed for about 10–15 min. The exercise comprised 30-min muscle strength and stretching exercises that included aerobic exercise, and the intensity was adjusted in stages by dividing the exercise into three cycles. The exercise intensity was incrementally increased at a maximum heart rate (HRmax) of 40–65% in Cycle 1 (Weeks 1–4), 55–60% HRmax in Cycle 2 (Weeks 5–8); and 60–70% HRmax in Cycle 3 (Weeks 9–12).

The *SaBang-DolGi* walking exercise program was designed by combining the principle of the *SaBang-ChiGi* of Korean dance in the early 2000s with a walking exercise; participants can improve their physical balance and strength through *SaBang-DolGi* based on walking in the same place. In this complex exercise that combines aerobics, strength, stretching, balance and jumping, participants start at the front with a powerful command and move 90° clockwise or in the opposite direction to conduct a pre-configured program (Figure 3). Remembering the sequence of movements in addition to performing musculoskeletal exercises is performed in an attempt to increase concentration and cognitive power [42].

The basic motion is walking in the same place so that the big toe touches the ground first. Another motion involves spreading both legs as wide as shoulder width, as shown in Figure 4. Based on the proposed basic movement, participants repeat the combined sequence of each motion 10 times in the same place, switching to four directions: east, west, north and south. Regarding the features of this program, first, a direction change accounting for 20–50% of the walking activity in everyday life [43] is a difficult motion [44] because the center of the body must be moved while maintaining dynamic stability. Avoiding obstacles while walking is an essential factor in this movement, so it is applied as an important factor in the exercise methods. The intensity of the exercise can be adjusted from low intensity to high intensity according to the subject and health condition, and it is designed to be easily accessible by anyone regardless of their age or gender. It is possible to continuously participate in different programs without becoming bored after a certain amount of time, and *SaBang-DolGi* walking has the advantage of allowing those with difficulties walking to participate in the program while sitting in a chair.

Second, chanting slogans relieves stress and helps participants to perform a certain motion in unity. Chanting “Let’s live healthily! Let’s live happily! Let’s live with gratitude!” at the beginning or end of the exercise allows participants to conceive of the body and mind as one and train them organically, thus strengthening positive thinking and self-confidence, which is the philosophy of this exercise.

During the first week, which is the adaptation process, participants were instructed to understand and familiarize themselves with the correct posture and motion of *SaBang-DolGi* walking. All participants participated at the same time in the program shown in Table 3.

### 2.4. Statistical Analysis

In this study, to confirm the effectiveness of the *SaBang-DolGi* walking exercise program, a two-way variance analysis was conducted by processing the collected data using the SPSS 23.0 (IBM, Armonk, NY, USA) for Windows statistical package program. In addition, the difference between measurements before and after participation in the exercise was examined to see the correlation between variables, and the difference between the changes was analyzed. Statistical significance levels were set to *p* < 0.05, *p* < 0.01, and *p* < 0.001.

## 3. Results

Based on the physical health test of the two groups, the differences in the body composition skeletal muscle mass (*p* = 0.001), BMI (*p* = 0.001), body fat percentage (*p* = 0.001), waist circumference (*p* = 0.001), hip circumference (*p* = 0.047), grip strength (*p* = 0.001), flexibility (*p* = 0.001) and abdominal muscle strength (*p* = 0.001) were found to be statistically significant in terms of both the groups and the period of measurement, as shown in Table 4.

Table 5 shows the results of mental health tests for the two groups in the domains of emotion, adaption to reality and others. In the emotional domain, the interactive effectiveness between the group and the period was found to be statistically significant for depression (*p* = 0.013), phobic anxiety (*p* = 0.016) and agoraphobia (*p* = 0.044). In the adaptation to reality, there was no statistical significance. In other domains, the interactive effectiveness between the group and the period were found to be statistically significant for sleep (*p* = 0.050), stress vulnerability (*p* = 0.020) and low self-regulation (*p* = 0.029).

Table 6 shows the correlation of the differences in physical and mental health before and after participation. The difference in skeletal muscle mass vs. obsessive–compulsive personality traits showed a positive (+) correlation of r = 0.320 (*p* < 0.05), and the difference in body fat percentage vs. obsessive–compulsion showed a positive (+) correlation of r = 0.345 (*p* < 0.05), indicating that, if the difference in skeletal muscle mass vs. the difference in body fat percentage is large, the difference in the obsessive–compulsive personality trait becomes greater.

The difference in grip strength vs. the difference in depression was r = 0-.391 (*p* < 0.05), the difference in anxiety was r = −0.374 (*p* < 0.05), the difference in phobic anxiety was r = −0.319 (*p* < 0.05), the difference in stress vulnerability was r = −0.346 (*p* < 0.05) and the difference in self-regulation was r = −0.392 (*p* < 0.05), showing a negative (−) correlation. These results indicate that, if the difference in grip strength is large, the scores for the difference in depression, anxiety, phobic anxiety, stress vulnerability and low self-regulation decrease.

The difference in flexibility vs. the difference in depression was r = −0.313 (*p* < 0.05), the difference in sleep problems was r = −0.400 (*p* < 0.05) and the difference in stress vulnerability was r = −0.348 (*p* < 0.05), showing a negative (−) correlation. These results indicate that, if the difference in flexibility is large, the scores for the difference in depression, sleep problems and vulnerability are small.

The difference in waist circumference vs. manic episodes was r = 0.453 (*p* < 0.01) and the difference in low-self regulation was r = 0.314 (*p* < 0.05), showing a positive (+) correlation. These results indicate that, if the difference in waist circumference is large, the scores for the difference in manic episodes and low self-regulation are large.

In other words, the difference in skeletal muscle mass vs. the difference in obsessive–compulsive personal traits, the difference in body fat percentage vs. the difference in obsessive–compulsion, the difference in waist circumference vs. the difference in manic episodes and the difference in low self-regulation showed a positive (+) correlation, whereas the difference in grip strength vs. the difference in depression, anxiety, phobic anxiety, stress vulnerability and low self-regulation and the difference in flexibility vs. the difference in depression, sleep problems and stress vulnerability showed a negative (−) correlation.

## 4. Discussion

In the current study, we analyzed the effectiveness of a 12-week *SaBang-DolGi* walking exercise program on the physical and mental health and changes in 40 menopausal women aged 50–65 after participating in the program. Women experience a variety of individual symptoms, including menopausal symptoms. The most prominent feature of many changes in physical function is reduced physical strength, and hormonal imbalance and decreased estrogen secretion make it difficult for women experiencing menopause to control their emotions. As exercise therapy has no side effects, costs little and helps to improve physical health as well as mental health, this study applied the *SaBang-DolGi* walking exercise program as an intervention and analyzed its results.

As a result of this study, it was found that BMI, body fat percentage and waist circumference decreased after exercise in the group of post-menopausal women and that their skeletal muscle mass increased, showing statistically significant changes. In general, the cause of weight gain during the post-menopausal period is the rapid decrease of hormones from the ovary in conjunction with the aging process, and the accumulation of body fat is promoted; in particular, abdominal fat cells increase in number [45]. Therefore, exercise is an effective way for women to reduce fat after the menopause, along with altering their diet and drug use [46]. Looking at the results of prior studies on the effectiveness of exercise mediation on post-menopausal women, weight, body fat percentage and BMI decreased in the obese women who participated in the 12-week circulation exercise program without using any tools that combined aerobic and resistance exercise [47], and the exercise program showed a positive effect in the group participating in complex exercise programs [48]. It was also reported that the exercise combined with aerobic exercise and resistance exercise increased energy consumption, leading to a positive effect in terms of changes in body composition [49]. These results are consistent with the results of studies reporting that aerobic exercise induces a positive effect on body fat reduction [50] and that resistance exercise improves muscle mass and decreases body fat mass [51] as a mechanism affecting body composition. Therefore, it is considered that the *SaBang-Dol G* walking exercise program, in the form of complex exercise, was effective in terms of reducing BMI, weight and waist circumference and increasing skeletal muscle mass. In particular, physical changes due to weight gain exacerbate the symptoms of serious diseases [52], thus it is necessary to make efforts to prevent this rather than overlooking the change in health status as a natural phenomenon caused by the menopause.

Participation in the *SaBang-DolGi* walking program showed statistically significant changes in grip strength, abdominal muscle strength and flexibility. Given the fact that the prevalence of muscular dystrophy in post-menopausal women is 10–40% [53], regular exercise to strengthen muscle strength and endurance in the post-menopausal period is very important [54]. In particular, it has been reported that grip strength, as an indicator of muscle strength reduction [55], can limit physical functions and cause physical strength reduction [56], and it also serves as an important factor that can predict mortality [57]. Therefore, it is considered that the most effective approach is to prevent muscle loss by increasing grip strength [58]. In prior studies on elderly people, it has been reported that grip strength was associated with cognitive ability and emotions as well as physical functions [59,60], indicating that maintaining grip strength is an important indicator not only for menopausal women but also to encourage an active and healthy life for the elderly. In addition, it has been reported that an increase in flexibility to prevent musculoskeletal system damage prevents the risk of muscle damage in daily life, thus promoting athletic performance [61].

As it has been proven that weakness inevitably leads to aging and strength loss can be improved by doing exercise through muscle strength-training programs [62,63], making habitual efforts to reduce health risk factors from middle-age is very important for successful aging.

Physical changes caused by menopause act as a stress in women, leading to mental illness [64]. As the World Health Organization (WHO) [65] also states depression as the biggest cause of disability worldwide, the symptoms of depression, which is now a public health issue around the world, are closely associated with quality of life and health risks such as cardiovascular diseases, etc. In addition, anxiety, one of the most common mental health disorders, affects concentration, sleep and daily work performance, which is closely associated with low physical activity [66]. As a result of this study, it was found that depression, anxiety, phobic anxiety, agoraphobia, sleep problems, stress vulnerability and low self-regulation in the sub-domain of mental health were significantly improved after exercise. These results are consistent with the results of prior studies on the positive effectiveness of exercise on mental health [8,62,63,67,68]. In particular, as it is associated with psychological changes [69,70,71], physical activity can improve general menopausal symptoms and mental health, such as mood disorders and insomnia [10]. In particular, it was reported that walking positively relieved depression and anxiety [72], and that a combination of aerobic and resistance exercise positively affected overall well-being and quality of life, e.g. by reducing anxiety and depression [68]. Moreover, regular exercise was closely associated with the relief of depression and anxiety [66]. Therefore, exercise can be effective for improving mental health vulnerabilities and may be viable adjuvant therapy for improving psychological stability. Therefore, the fun factor of the *SaBang-DolGi* walking program used in this study is considered to provide an escape for those who are mentally suffering.

Lastly, looking at the results of the analysis of the differences in changes between before and after participating in the exercise, the difference in skeletal muscle mass vs. the difference in obsessive–compulsive personality traits was r = 0.320 (*p* < 0.05), the difference in body fat percentage vs. the difference in obsessive–compulsion was r = 0.345 (*p* < 0.05), the difference in waist circumference vs. the difference in manic episodes was r = 0.453 (*p* < 0.05) and the difference in low self-regulation was r = 0.314 (*p* < 0.05), showing a positive (+) correlation. Furthermore, the difference in grip strength vs. the difference in depression was r = −0.391 (*p* < 0.05), the difference in anxiety was r = −0.374 (*p* < 0.05), the difference in phobic anxiety was r = −0.319 (*p* < 0.05), the difference in stress vulnerability was r = 0−0.0346 (*p* < 0.05), the difference in low self-regulation was r = −0.392 (*p* < 0.05), the difference in flexibility vs. the difference in depression was r = −0.313 (*p* < 0.05), the difference in sleep problems was r = −0.400 (*p* < 0.05) and the difference in stress vulnerability was r = −0.348 (*p* < 0.05), showing a negative(−) correlation.

Being physically active improves mental and musculoskeletal health, muscle strength and joint flexibility, and regular physical activity prevents fat tissue accumulation [68]. These results are consistent with the results of studies of the WHO [65] and NICE [73] reporting that regular physical activity and exercise have a positive effect on mental health. In prior cross-sectional studies, it was reported that increased muscle strength significantly reduced the risk of depressive symptoms, and this association consistently appeared in the community-based environment [74,75,76,77,78], which confirmed the relationship. As a result, *SaBang-DolGi* walking can help participants to manage and reduce the number of uncomfortable symptoms during the menopause by providing an opportunity to immediately walk in the same place without moving. The habit of walking in the same place will also play an important role in reducing sedentary activities, making the menopausal transition easier.

In summary, it was confirmed that the *SaBang-DolGi* walking exercise program is an effective exercise mediation for the physical and mental health of post-menopausal women because it can maintain the diversity, fun and persistence of exercise. To reduce depression and anxiety symptoms in mental health and counteract the reduced physical strength caused by the menopause, it would be helpful to encourage active participation in complex exercises, including resistance exercises, and, to relieve menopausal symptoms and promote health, it would be necessary to encourage the habit of doing exercise more than three times a week on a regular basis. In addition, given that the physical and mental health problems of menopausal women can affect the composition of a healthy society, spreading to the families and society to which they belong and extending beyond the quality of life of individuals, social concerns are also relevant.

## 5. Conclusions

The results of this study indicate that the *SaBang-DolGi* walking exercise program has a positive effectiveness on the physical and mental health of menopausal women. Moreover, the proposed protocol is associated with variables of physical health (body composition, flexibility, grip strength and abdominal muscle strength) and variables of mental health (depression, anxiety, phobic anxiety, agoraphobia, sleep problems, stress vulnerability and low self-regulation). Therefore, to continue social health activities without being exposed to disease after the menopause, it is necessary to maintain physical and mental health from the perspective of health–sociology. In addition, as the difficulties in participating in group exercise after the outbreak of Covid-19 has led to a decrease in physical activity, it is considered that this exercise program, which can be performed at home alone, can be disseminated as a means of healthy sociological exercise, and the results of this study are expected to serve as the basic data for this purpose.

Since the number of women in the program was small and it was a short-term study, there are limitations to the results of this research. As physical and mental health has recently been understood to be affected by the active social activities of women, there is a need to analyze not only menopausal women but also changes in quality of life, exercise intensity and continuous exercise practice. Additional exclusionary criteria were implemented which excluded women with surgical menopause or women using hormone therapy—that is, samples of women likely to experience the most severe symptoms—warranting further investigation of these clinically vulnerable groups. Physical activity interventions of longer durations would also be desirable for the elicitation of more pronounced and longer-lasting intervention effects. Thus, future studies should employ larger and more representative samples to yield more reliable and generalizable results.

## Figures and Tables

**Figure 1 ijerph-17-06935-f001:**
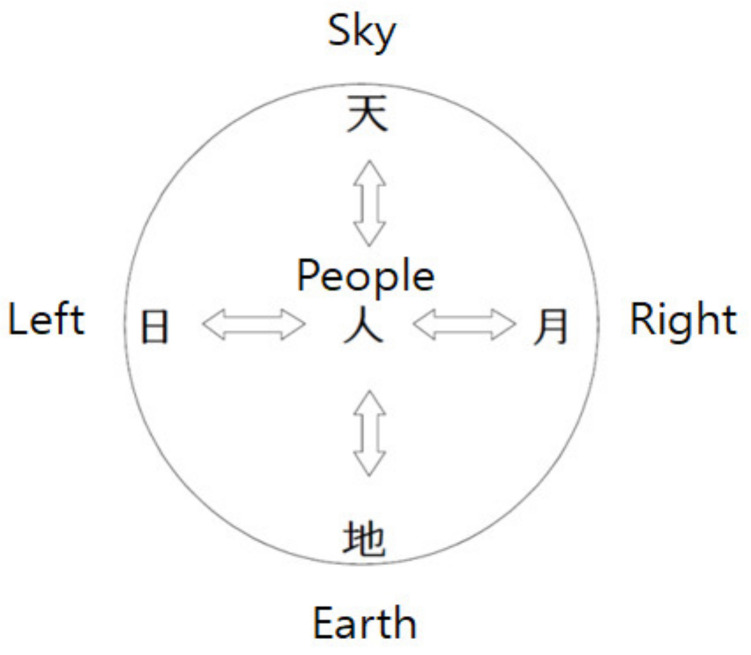
天地人日月圖: The meaning of the epidemiological spatial structure of Korea Dance, (Choen Ji In Il Wol Do: spatial structure—sky (天), earth (地) and people (人), temporal structure: left (日), right (月); interacting in a state of harmony and balance, creating movement of motion) states that humans who respond to the sky above and the earth below interact with heaven, the earth and all directions.

**Figure 2 ijerph-17-06935-f002:**
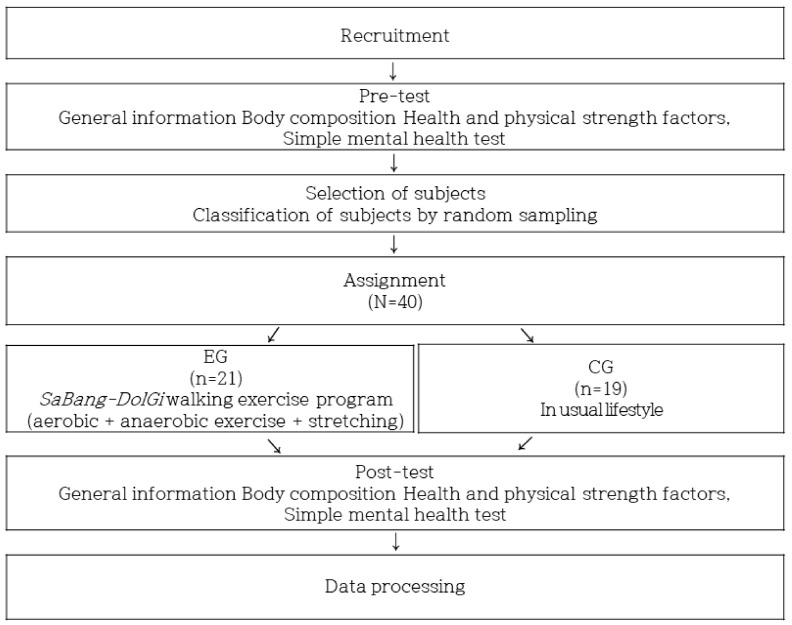
Participant allocation (flow diagram of the consolidated standards for the reporting of trials).

**Figure 3 ijerph-17-06935-f003:**
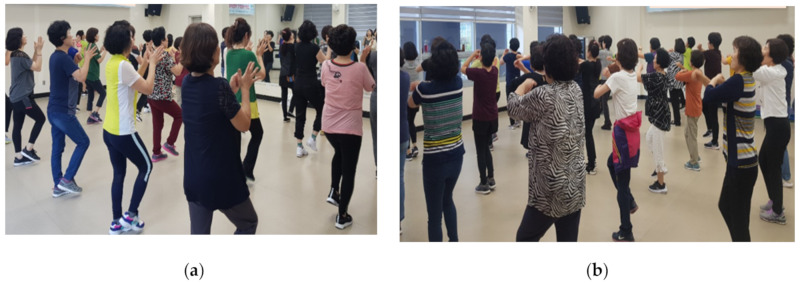
Exercises performed: (**a**) walking while clapping; and (**b**) walking while turning the wrist.

**Figure 4 ijerph-17-06935-f004:**
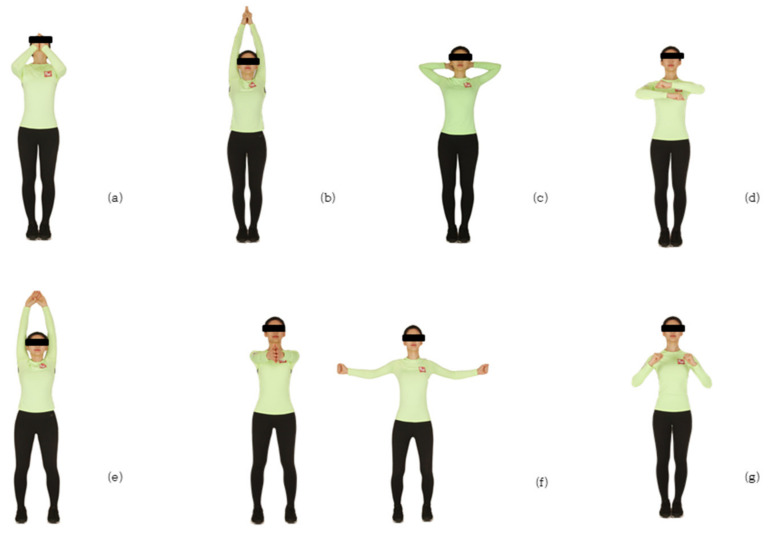
Exercises performed: (**a**) walking with the chin; (**b**) walking while praying; (**c**) walking while stretching shoulders; (**d**) walking while turning the wrist; (**e**) erector spinal muscle; (**f**) stretching the chest; and (**g**) jjamjjam walking.

**Table 1 ijerph-17-06935-t001:** Physical characteristics of participants in the study.

Group	Age (years)	Height (cm)	Body Weight (kg)	BMI (kg/m^2^)	Fat (%)	Waist Circumference (cm)
EG (*N* = 21)	59.38	157.94	58.27	23.31	18.94	80.27
	±3.76	±5.47	±7.37	±2.79	±5.27	±7.08
CG (*N* = 19)	58.21	156.27	61.58	25.17	22.07	84.35
	±3.99	±4.97	±10.58	±3.77	±6.17	±7.79

Mean ± standard deviation (SD); EG, experimental group; CG, control group.

**Table 2 ijerph-17-06935-t002:** Validity and reliability of mental health test.

Domain	Characteristic	1	2	3	Communalities	Cronbach’s α
Emotion	Social Desirability	0.898	−0.209	0.079	0.687	0.894
	Inconsistency	0.870	0.334	0.258	0.771	
	Depression	0.857	0.342	0.249	0.752	
	Anxiety	0.819	0.493	0.072	0.764	
	Phobic Anxiety	0.796	0.433	0.201	0.646	
	Panic Attack	0.783	0.119	0.126	0.630	
	Agoraphobia	0.771	−0.012	0.525	0.674	
	Obsessive-Compulsive	0.758	−0.035	−0.003	0.655	
	Obsession	0.739	0.393	0.272	0.599	
	Obsessive-Compulsive Personality Trait	0.683	0.106	0.024	0.791	
	Posttraumatic Stress Disorder	0.672	0.336	0.445	0.803	
	Aggression	0.650	0.116	0.118	0.699	
	Somatization	0.642	0.457	0.374	0.642	
Others	Suicide	0.605	0.710	0.122	0.753	0.863
	Addiction	0.363	0.648	0.077	0.638	
	Sleep Problem	0.214	0.634	0.018	0.601	
	Stress Vulnerability	0.268	−0.070	0.702	0.670	
	Interpersonal Sensitivity	0.181	0.209	0.692	0.656	
	Self-Regulation Problem	0.037	0.223	0.607	0.619	
Adaptation to reality	Manic episode	0.105	0.729	0.322	0.646	0.697
	Paranoia	0.419	0.718	0.162	0.718	
	Schizophrenia	0.555	0.714	0.147	0.840	
Characteristic value		10.299	1.796	1.560		
Explanation variance		46.813	8.165	7.089		
Cumulative variance		46.813	54.978	62.067		
KMO = 0.719 Bartlett’ Test of Sphericity = 1110.059 (df = 231, sig. = 0.000)						

1, emotion; 2, others; 3, adaptation to reality. KMO, kaiser meyer olkin.

**Table 3 ijerph-17-06935-t003:** *SaBang-DolGi* walking exercise program.

Item	Description	Intensity	Remark	Frequency
Warn-up exercise (Aerobics) 10 min	*1W, **3W 1. Walking while turning the wrist. 2. Tapping the head, tapping the chest, clapping.	RPE 6–8	*SaBang-DolGi* (one or two turns) Healthy kinship Maintaining relationships	3 sessions/week
Muscle strengthening 10 min	Stretching the chest, erector spinal muscle, stretching over the shoulders. Cross PT up, cross PT down, down walking. 10 new programs for each session.	HR max 40–60% RPE 9–11 (Weeks 1–6)	*SaBang-DolGi* (One or two turns)	
		HR max 60–80% RPE 12–16 (Weeks 7–12)		
Complex exercise 25 min	***1B1W 1. Stretching the chest, walking with the chin. 2. Turning shoulders, tapping the abdomen. ****1B2W 1. Erector spinal muscle, praying, stretching shoulders. 2. Stretching the chest, jjamjjam walking_right, left. 3. Stretching over shoulders, walking while turning the wrist_ in or outward.	HR max 40–60% RPE 9–11 (Weeks 1–6)	Muscle strengthening motion + walking motion + stretching *SaBang-DolGi* (one or two turns)	
		HR max 60–80% RPE 12–16 (Weeks 7–12)		
Warm-down exercise 15 min	**3W 1. Jjamjjam, shaking the wrist, clapping. 2. Clapping and walking in the place. 3. Walking while tapping the shoulder of a person in front (Train walking). **3W and chanting balance slogans.	RPE 7-9	*SaBang-DolGi* (one or two turns) Pursuing fun Chanting balance slogans. “Let’s live healthily! Let’s live happily! Let’s live with gratitude!”	

W, walking; B, balance; HR, heart rate; RPE, rating of perceived exertion; PT, physical training and personal training; *1W, one walking motion; **3W, three walking motion; ***1B1W, one balance motion and one walking motion; ****1B2W, one balance motion and two walking motion.

**Table 4 ijerph-17-06935-t004:** Changes in physical health after participating in the *SaBang-DolGi* walking exercise program.

Domain	Characteristic	Period	Exp. (*N* = 21)	Con. (*N* = 19)	ANOVA (*p*)		
			M ± SD	M ± SD	G	T	G × T
Physical health	Skeletal muscle mass	Pre	21.13 ± 2.17	21.41 ± 3.82	0.971	0.588	0.001 ***
		Post	21.47 ± 2.26	21.13 ± 2.90			
	BMI	Pre	23.31 ± 2.79	25.18 ± 3.77	0.035 *	0.689	0.001 ***
		Post	22.90 ± 2.64	25.55 ± 3.82			
	Body fat percentage	Pre	32.09 ± 5.62	35.39 ± 4.33	0.020 *	0.124	0.001 ***
		Post	31.35 ± 5.75	35.84 ± 4.24			
	Waist circumference	Pre	80.27 ± 7.08	84.35 ± 7.79	0.020 *	0.957	0.001 ***
		Post	78.63 ± 7.01	86.03 ± 8.24			
	Hip circumference	Pre	95.05 ± 5.17	98.69 ± 7.42	0.109	0.010 **	0.047 *
		Post	95.81 ± 5.44	98.79 ± 7.43			
	Grip strength	Pre	21.69 ± 3.02	20.83 ± 5.72	0.107	0.404	0.001 ***
		Post	23.35 ± 3.27	19.60 ± 5.53			
	Abdominal muscle strength	Pre	14.62 ± 2.82	14.26 ± 2.42	0.001 ***	0.001 ***	0.001 ***
		Post	21.19 ± 2.60	14.26 ± 2.21			
	Flexibility	Pre	14.09 ± 8.10	12.18 ± 7.53	0.043 *	0.001 ***	0.001 ***
		Post	18.79 ± 7.59	10.30 ± 8.48			

All data represent mean ± standard deviation. Exp., experimental group; Con., control group; Pre., pretest; Post., posttest; M., mean; SD., standard deviation; G., group; T., time; G × T., groups × time. * *p* < 0.05; ** *p* < 0.01; *** *p* < 0.001.

**Table 5 ijerph-17-06935-t005:** Changes in mental health after participating in the *SaBang-Dol Gi* walking exercise program.

Domain	Characteristic	Period	Exp. (*N* = 21)	Con. (*N* = 19)	ANOVA (*p*)		
			M ± SD	M ± SD	G	T	G × T
Emotion	Social Desirability	Pre	54.00 ± 12.67	53.42 ± 6.40	0.570	0.872	0.160
		Post	51.90 ± 13.62	56.05 ± 9.84			
	Inconsistency	Pre	50.05 ± 10.66	50.89 ± 13.51	0.971	0.461	0.776
		Post	48.86 ± 9.55	48.21 ± 14.08			
	Depression	Pre	48.38 ± 7.03	46.89 ± 8.56	0.963	0.001 ***	0.013 *
		Post	43.57 ± 7.02	45.84 ± 8.94			
	Anxiety	Pre	51.71 ± 8.32	48.84 ± 10.48	0.650	0.037 *	0.086
		Post	48.10 ± 6.56	48.47 ± 10.68			
	Phobic Anxiety	Pre	49.19 ± 6.18	48.05 ± 8.65	0.798	0.072	0.016 *
		Post	46.10 ± 4.66	48.53 ± 12.03			
	Panic Attack	Pre	51.43 ± 9.07	47.53 ± 6.23	0.328	0.038 *	0.090
		Post	46.90 ± 5.07	47.05 ± 7.03			
	Agoraphobia	Pre	47.38 ± 5.76	49.00 ± 12.91	0.368	0.856	0.044 *
		Post	46.00 ± 5.19	50.16 ± 14.32			
	Obsessive-Compulsive	Pre	49.76 ± 9.47	49.74 ± 11.52	0.708	0.094	0.352
		Post	46.52 ± 5.46	48.79 ± 13.01			
	Obsession	Pre	52.00 ± 10.12	46.63 ± 10.91	0.315	0.323	0.081
		Post	48.29 ± 8.18	47.68 ± 11.37			
	Obsessive-Compulsive Personality Trait	Pre	46.90 ± 9.64	53.47 ± 14.60	0.109	0.119	0.515
		Post	45.52 ± 5.62	50.16 ± 15.17			
	Posttraumatic Stress Disorder	Pre	49.05 ± 10.97	49.37 ± 14.24	0.863	0.126	0.888
		Post	46.14 ± 8.34	46.95 ± 12.17			
	Aggression	Pre	46.52 ± 6.01	45.58 ± 8.51	0.834	0.240	0.621
		Post	44.86 ± 4.45	44.89 ± 10.11			
	Somatization	Pre	51.00 ± 10.97	46.53 ± 7.21	0.150	0.001 ***	0.496
		Post	46.14 ± 9.33	43.21 ± 6.12			
Adaptation to reality	Manic episode	Pre	51.71 ± 10.12	53.42 ± 15.99	0.484	0.571	0.596
		Post	50.14 ± 8.84	53.37 ± 11.88			
	Paranoia	Pre	46.38 ± 7.67	45.68 ± 8.16	0.904	0.904	0.552
		Post	45.86 ± 9.18	46.47 ± 9.82			
	Schizophrenia	Pre	48.81 ± 7.75	50.84 ± 9.40	0.338	0.327	0.703
		Post	47.14 ± 6.53	50.11 ± 11.75			
Others	Suicide	Pre	45.76 ± 6.19	44.00 ± 5.89	0.781	0.914	0.242
		Post	44.67 ± 7.69	45.31 ± 8.14			
	Addiction	Pre	46.76 ± 2.41	50.89 ± 0.00	0.466	0.274	0.655
		Post	47.10 ± 3.63	46.79 ± 3.44			
	Sleep Problem	Pre	53.95 ± 8.74	49.58 ± 10.21	0.410	0.144	0.050 *
		Post	50.57 ± 8.98	50.05 ± 11.13			
	Stress Vulnerability	Pre	51.71 ± 8.32	48.84 ± 10.48	0.488	0.026 *	0.020 *
		Post	48.10 ± 6.56	48.47 ± 10.68			
	Interpersonal Sensitivity	Pre	52.24 ± 7.15	49.42 ± 10.83	0.796	0.318	0.072
		Post	49.00 ± 5.69	50.37 ± 12.98			
	Self-Regulation Problem	Pre	51.19 ± 7.33	46.05 ± 8.79	0.265	0.005 **	0.029 *
		Post	45.86 ± 7.30	45.32 ± 10.58			

All data represent mean ± standard deviation. Exp., experimental group; Con., control group; Pre, pretest; Post, posttest; M, mean; SD, standard deviation; G, group; T, time; G × T, groups × time. * *p* < 0.05; ** *p* <0.01; *** *p* < 0.001.

**Table 6 ijerph-17-06935-t006:** Correlations according to the changes in physical and mental health before and after participating in the exercise.

	1	2	3	4	5	6	7	8	9	10	11	12	13	14	15	16	17	18	19	20	21	22	23	24	25	26	27
**1**	1																										
**2**	−0.0208	1																									
**3**	−0.0473 **	0.627 **	1																								
**4**	0.431 **	−0.0519 **	−0.0525 **	1																							
**5**	0.598 **	−0.0539 **	−0.0546 **	0.586 **	1																						
**6**	−0.0316 *	0.476 **	−0.0501 **	−0.0218	−0.0459 **	1																					
**7**	0.087	−0.0310	−0.0176	0.027	0.177	−0.0250	1																				
**8**	−0.0289	0.233	0.310	−0.0391 *	−0.0313 *	0.114	−0.0232	1																			
**9**	−0.0115	0.179	0.177	−0.0374 *	−0.0304	0.148	0.025	0.606 **	1																		
**10**	−0.0202	0.196	0.285	−0.0319 *	−0.0222	0.027	−0.0097	0.397 *	0.302	1																	
**11**	−0.0183	0.108	0.266	−0.0280	−0.0103	−0.0032	−0.0090	0.373 *	0.263	0.863 **	1																
**12**	−0.0061	0.251	0.135	−0.0178	−0.0287	0.137	−0.0064	0.171	0.166	0.562 **	0.074	1															
**13**	0.143	0.292	0.271	−0.0125	−0.0012	0.017	0.054	0.229	0.264	0.075	0.181	−0.0102	1														
**14**	−0.0073	0.276	0.345 *	−0.0246	−0.0121	0.080	−0.0031	0.163	0.305	0.190	0.235	0.034	0.750 **	1													
**15**	0.320 *	0.128	0.021	0.093	0.136	−0.0090	0.129	0.152	0.061	−0.0118	−0.0011	−0.0191	0.712 **	0.073	1												
**16**	0.034	−0.0056	−0.0033	−0.0008	0.036	0.021	−0.0123	0.373 *	0.246	0.133	0.107	0.104	0.301	0.267	0.181	1											
**17**	−0.0039	0.143	0.066	−0.0089	−0.0080	−0.0058	0.118	0.323 *	0.548 **	0.479 **	0.466 **	0.162	0.311	0.159	0.285	0.175	1										
**18**	−0.0030	0.114	0.252	−0.0227	−0.0111	−0.0155	0.178	0.300	0.307	0.467 **	0.466 **	0.148	0.392*	0.349*	0.211	0.318 *	0.547 **	1									
**19**	0.116	0.103	0.035	0.016	0.060	0.453 **	−0.0311	0.146	0.268	−0.0025	0.059	−0.0114	0.077	0.125	−0.0038	0.290	0.180	−0.0083	1								
**20**	−0.0085	0.039	0.051	−0.0257	−0.0090	−0.0074	0.061	0.322 *	0.491 **	0.328 *	0.354 *	0.050	0.150	0.167	0.046	−0.0031	0.598 **	0.195	0.238	1							
**21**	0.023	−0.0096	0.144	−0.0104	−0.0087	0.041	0.177	0.305	0.248	0.213	0.272	−0.0017	0.192	0.399*	−0.0132	0.057	−0.0052	0.144	0.031	0.365 *	1						
**22**	−0.0175	0.138	0.201	−0.0123	−0.0266	0.019	−0.0205	0.677 **	0.379 *	0.139	0.039	0.192	−0.0014	−0.0204	0.194	0.127	0.271	0.065	−0.0025	0.215	0.142	1					
**23**	−0.0060	0.014	0.062	0.027	−0.0040	0.120	−0.0150	0.424 **	0.089	−0.0024	−0.0112	0.121	0.013	−0.0179	0.208	0.333 *	−0.0034	−0.0033	0.136	−0.0131	−0.0035	0.565 **	1				
**24**	−0.0247	0.232	0.078	−0.0216	−0.0400 *	0.302	0.064	0.220	0.496 **	−0.0168	−0.0156	−0.0079	0.051	0.074	−0.0012	−0.0116	0.054	−0.0034	−0.0053	0.173	0.164	0.126	−0.0046	1			
**25**	−0.0024	0.297	0.139	−0.0346 *	−0.0348 *	0.302	0.027	0.637 **	0.571 **	0.348 *	0.266	0.288	0.256	0.237	0.123	0.369 *	0.352 *	0.313 *	0.341 *	0.327 *	0.358 *	0.447 **	0.288	0.145	1		
**26**	0.008	0.255	0.037	−0.0211	−0.0286	0.201	0.082	0.468 **	0.433 **	0.321 *	0.149	0.436 **	0.186	0.178	0.083	0.449 **	0.292	0.267	0.265	0.163	0.124	0.293	0.273	−0.0038	0.866 **	1	
**27**	−0.0070	0.241	0.215	−0.0392 *	−0.0296	0.314 *	−0.0059	0.621 **	0.525 **	0.257	0.306	0.019	0.236	0.217	0.115	0.166	0.271	0.315*	0.315*	0.396*	0.507 **	0.489 **	0.213	0.287	0.814 **	0.420 **	1

* *p* < 0.05; ** *p* < 0.01. (1) difference in skeletal muscle mass; (2) difference in BMI; (3) difference in body fat percentage; (4) difference in grip strength; (5) difference in flexibility; (6) difference in waist circumference; (7) difference in hip circumference; (8) difference in depression; (9) difference in anxiety; (10) difference in phobic anxiety; (11) difference in panic attacks; (12) difference in agoraphobia; (13) difference in obsessive-compulsion; (14) difference in obsession; (15) difference in obsessive-compulsive personality trait; (16) difference in PTSD (Post-Traumatic Stress Disorder); (17) difference in panic attacks; (18) difference in somatization; (19) difference in manic episodes; (20) difference in paranoia; (21) difference in schizophrenia; (22) difference in suicide; (23) difference in addiction; (24) difference in sleep problems; (25) difference in stress vulnerability; (26) difference in interpersonal sensitivity; (27) difference in low self-regulation.

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
