# Peer review of "Effectiveness of *SaBang-DolGi* Walking Exercise Program on Physical and Mental Health of Menopausal Women"

_ijerph, 2020, doi:10.3390/ijerph17186935_

Round 1
Reviewer 1 Report
Effectiveness of Sa Bang Dol Gi walking exercise program on physical and mental health of middle-aged menopausal women
Thanks for the opportunity to review this interesting piece of work. If some of the concerns below are included, the paper will add a substantial value to the literature on women’s health. However, it needs a substantial language edition should the paper is accepted for publication.
Abstract
- The authors need to define ‘Sa Bang Dol Gi’ at the beginning, and I would prefer the term to be italic throughout as it is a locale.
- Line 18-22, Could the authors break the sentence in to two? Perhaps, the test and outcome may be one statement.
- Line 22-25, the authors need to concisely and clearly write the exposure, the outcomes and the mode of analysis----descriptive and inferential (i.e. correlation) analyses techniques.
Introduction
- First of all, could the authors review literature and report summary of the effect of ‘Sa Bang Dol Gi’ on health and health outcomes in the general population in Korea. Additionally, the mechanism of action and potential confounders should be stated. For example, do people with depression feel better due to the exercise or the ‘chanting or music’ which accompany the Sa Bang Dol Gi. There are several literature showing classical music have a positive impact on relieving mental illness.
- IN relation to this, how can the authors argue that investigating the impact of Sa Bang Dol Gi on mental illness (which perhaps may exclude people with physical disability) as opposed to investigating the impact of Korean classical music on mental illness (which can include those people with physical disability, except deafness)?
- And the gap has to be clearly defined, particularly, the absence of data on the problem among those population in Korea using robust methods such as the one used in this paper.
- How do the authors protect transmission infectious diseases such as Tuberculosis, COVID-19, etc…
- There are some editorial issues, as described below ere in the introduction section.
- Line 49-53, Could the authors provide citation?
- Line 59-61, ditto- we need citation/s that complement/s exercise therapy’s positive impact on menopausal symptoms, including mental health symptoms, and physical symptoms separately.
- Line 64….reducing what [11,12]
- Line 85-86,,,,helping help increase
- Line 93: title for figure 1 has to be translated in to English
- Line 108-9: cite the few studies.
Materials and methods
- Line 114: Why do not the author calculate a sample size?
- Line 160: could the authors add the validity and reliability of the tool?
- Line 221, analysis: what don’t the authors apply advanced inferential statistical analysis to quantify the magnitude of difference before and after the exercise?
Results
- Table 5 can be appended.
- I was wondering if the authors also measure the validity and reliability of the tool in their population
- Could the authors know (and report) for study participants who gained a positive outcome in physical AND mental health? Any composite score to magnify the importance of ‘Sa Bang Dol Gi’
- Have the authors assessed the impact of age on their outcome? 50s vs 60s?
Discussion
- Line 10, what is BMD? Is that BMI?
- Line 15 ….the only way to for….check the statement
- Any future research recommendations?
- Any limitation of the study?
Author Response
Response to Reviewer 1 Comments
Appreciate for your insightful and useful comments.
We’ve revised to reflect your comment.
Abstract
Point 1: The authors need to define ‘Sa Bang Dol Gi’ at the beginning, and I would prefer the term to be italic throughout as it is a locale.
Response 1:
Modified to italic font ‘SaBang-DolGi’.
Thank you
Point 2: Could the authors break the sentence in to two? Perhaps, the test and outcome may be one statement.
Response 2:
As requested, the contents have been modified and supplemented.
- Change completed (line 18-23).
- (line 18-23)The subjects participating in this study were classified into the experimental(exercise) group (n=21) and the control group (n=19), and physical (grip, muscle, and endurance) test and mental health test (simple mental health test II) were conducted using questionnaires with the aim of examining subjects’ physical and mental health before and after exercise. --> Material and methods The participants comprised 40 women aged 50–65 years who were divided into two randomly selected groups in training sessions (exercising group, n = 21 and control group, n = 19). A physical (grip, muscle and endurance) test and mental health test (simple mental health test II) were conducted using questionnaires with the aim of examining subjects’ physical and mental health before and after exercise.
Thank you
Point 3: The authors need to concisely and clearly write the exposure, the outcomes and the mode of analysis----descriptive and inferential (i.e. correlation) analyses techniques.
Response 3:
As requested, the contents have been modified and supplemented.
- Change completed (line 23-26).
- (line 23-26)In order to examine the effectiveness of Sa Bang Dol Gi walking exercise program, the differences in changes between before and after the application of the program were compared and analyzed. In addition, the correlation between variables were analyzed by examining the differences in changes between before and after participation in the exercise program. --> Results: After the intervention, the participants experienced positive changes in the physical dimension, with significant enhancements particularly in mental well-being and menopause-related health and subdomains. Controlled and regular exercise for 12 weeks was significantly correlated with a positive change in vitality and mental health.
Thank you
Introduction
Point 4: First of all, could the authors review literature and report summary of the effect of ‘Sa Bang Dol Gi’ on health and health outcomes in the general population in Korea. Additionally, the mechanism of action and potential confounders should be stated. For example, do people with depression feel better due to the exercise or the ‘chanting or music’ which accompany the Sa Bang Dol Gi. There are several literature showing classical music have a positive impact on relieving mental illness. IN relation to this, how can the authors argue that investigating the impact of Sa Bang Dol Gi on mental illness (which perhaps may exclude people with physical disability) as opposed to investigating the impact of Korean classical music on mental illness (which can include those people with physical disability, except deafness)? And the gap has to be clearly defined, particularly, the absence of data on the problem among those population in Korea using robust methods such as the one used in this paper. How do the authors protect transmission infectious diseases such as Tuberculosis, COVID-19, etc…
Response 4:
As requested, the contents have been modified and supplemented.
- Insertion completed (line 85-90).
- (line 85-90) Rhythmic phrases and intensive repetitive movements have positive effects on bone mass, flexibility and mental health conditions such as depression, in a manner similar to lifestyle music therapy [30-34]. Repeated and intensive use of the full body, such as in SaBang-DolGi, has been reported to be important for the relocation of the motor field of the cerebral cortex [35]; furthermore, a reduction of depression has been reported due to the central nervous system inducing changes in the cerebral cortex level [36-37].
- When behavior is limited by infectious diseases such as COVID-19 and so on, we walk in place in the same way as "'SaBang-DolGi" and create a movement of motion.
Thank you
Point 5: There are some editorial issues, as described below ere in the introduction section.
Response 5:
As requested, the contents have been modified and supplemented.
- Insertion completed (line 47, 50, 53, 56-58, 80, 92-93, 99-101).
- (line 47) [5]Cha, S.H. ”The prevention and treatment of gynecological diseases for women” ISBN: , Garim Publishing House. 2003.
(line 50) [8]Stojanovska, L.; Apostolopoulos, V.; Polman, R.; Borkoles, E. To exercise, or, not to exercise, during menopause and beyond. Maturitas. 2014, 77(4), 318–23.
[9]Dalziel, K.; Segal, L.; Elley, C.R. Cost utility analysis of physical activity counselling in general practice. Aust N Z J Public Health. 2006, 30(1):57–63.
- (line 53) and reduce, --> reducing anxiety,
- (line 56-58) Physical activity during the menopausal transition and post-menopause period has been reported to improve mental health, prevent weight gain, increase bone mineral density and muscle mass and reduce and the risks of other diseases (e.g., cencer, diabetes, heart disease) [8, 20].
- (line 80) thus helping help increase --> thus it helps to increase
- (line 92-93) 天地人日月圖(Choen Ji In Il Wol Do): It means that humans who respond to the sky above and the earth below interact with heaven, earth, and all directions.
- (line 99-101) Nevertheless, the only research based on the program is the study of women's physical health factors, cardiovascular risk factors and quality of life after the menopause [38], and the static posture, walking variables and balancing ability of male college students [39].
Thank you
Materials and methods
Point 6: Line 114: Why do not the author calculate a sample size?
Line 160: could the authors add the validity and reliability of the tool?
Line 221, analysis: what don’t the authors apply advanced inferential statistical analysis to quantify the magnitude of difference before and after the exercise?
Response 6:
As requested, the contents have been modified and supplemented.
- Insertion completed (line 119-122, 187-190).
- (line 119-122) The sample size was calculated using the G-Power 3.1 program. The α level, test power, and effect size were set to 0.05, 0.80, and 0.50, respectively, so the minimum number of participants in this study was 34. However, in this study, 40 participants were recruited, which allowed for the potential dropout of some participants. --> The sample size was determined using the G*Power program, considering a significance level of 0.05 and a power of 0.80 to obtain an effect size of 0.5. It was determined that a sample size of 34 individuals was adequate, meaning that our sample satisfied the conditions for the recommended sample size.
- (line 187-190) The measurement of the factor analysis for the survey items used the emotional area with a Cronbach’s α of 0.894, other areas with a Cronbach’s α of 0.863, reality adaptation problems with a Cronbach’s α of 0.697; these values were mostly found to be 0.6 or higher, indicating reliability and validity.
Thank you
Results
Point 7: Table 5 can be appended.
I was wondering if the authors also measure the validity and reliability of the tool in their population
Could the authors know (and report) for study participants who gained a positive outcome in physical AND mental health? Any composite score to magnify the importance of ‘Sa Bang Dol Gi’
Have the authors assessed the impact of age on their outcome? 50s vs 60s?
Response 7:
As requested, the contents have been modified and supplemented.
- Insertion completed (Table 5).
- (Table 5) KMO = .719, Bartlett' Test of Sphericity = 1110.059(df=231, sig.=.000). The KMO test index was 0.719, which was larger than 0.7, and Barlett's spherical test index was 0.000 (<0.05).
- After controlling the age as a covariate, in the existing results, most of the covariates did not show any significant difference, indicating that the difference was due to the effect of the program.
Thank you
Discussion
Point 8: Line 10, what is BMD? Is that BMI?
Line 15 ….the only way to for….check the statement
Any future research recommendations?
Any limitation of the study?
Response8:
As requested, the contents have been modified and supplemented.
- Insertion completed (line 11, 16-17, 111-122).
- (line 11) BMD --> BMI
- (line 16-17) Therefore, exercise is the only way to for post-menopausal women to reduce boy fat, except for diet and medication [47]. --> Therefore, exercise is an effective way for women to reduce fat after the menopause, along with altering their diet and drug use [46].
- (line 111-122) Limitations --> Since the number of middle-aged women in the program was small and it was a short-term study, there are limitations to the results of this research. As physical and mental health has recently been understood to be affected by the active social activities of women, there is a need to analyze not only menopausal women but also middle-aged women's changes in quality of life, exercise intensity and continuous exercise practice. Middle-aged women need to analyze various factors related to the menopause, as it is a time when physical and mental changes occur. Additional exclusionary criteria were implemented which excluded women with surgical menopause or women using hormone therapy—that is, samples of women likely to experience the most severe symptoms—warranting further investigation of these clinically vulnerable groups. Physical activity interventions of longer durations would also be desirable for the elicitation of more pronounced and longer-lasting intervention effects. Thus, future studies should employ larger and more representative samples to yield more reliable and generalizable results.
Thank you
Reviewer 2 Report
ijerph-906153
The aim of the article is to investigate the effect of Sa Bang Dol Gi walking exercise on menopausal women. The authors found out that this kind of walking exercise increases the physical and mental health of middle-aged menopausal women particularly for those exposed to various stresses and depressions.
The topic of this manuscript is suggestive and falls within the scope of the journal. I would recommend the following revisions, to improve the article:
- All the text needs a language revision by a native English speaker person, in order to correct typos and style (repetitions, punctuation, grammar).
- Isoflavones and other nutraceutical compounds are widely used for their beneficial effects on menopausal complaints. Authors should deepen this aspect in the introduction, referring to PMID: 31466381; PMID: 32693763
- Figure 1: Authors should improve the caption and explain the symbols used.
- Materials and methods: please authors describe exclusion criteria clearly as these should be planned before the recruitment.
- Conclusions: Authors should change the word “Corona” with “Covid”
- References: Please authors complete the reference n°48
Author Response
Response to Reviewer 2 Comments
Appreciate for your insightful and useful comments.
We’ve revised to reflect your comment.
Point 1: All the text needs a language revision by a native English speaker person, in order to correct typos and style (repetitions, punctuation, grammar).
Response 1:
As requested, the contents have been modified and supplemented.
- We requested the MDPI editor to modify all texts.
Thank you
Point 2: Isoflavones and other nutraceutical compounds are widely used for their beneficial effects on menopausal complaints. Authors should deepen this aspect in the introduction, referring to PMID: 31466381; PMID: 32693763
Response 2:
As requested, the contents have been modified and supplemented.
- Insertion completed (line 47-48).
- In order to alleviate these symptoms, alternative therapies are widely used and show beneficial effects [6, 7]
Thank you
Point 3: Figure 1: Authors should improve the caption and explain the symbols used.
Response 3:
As requested, the contents have been modified and supplemented.
- Insertion completed (line 92-93).
- (line 92-93) Figure 1. 天地人日月圖 (Choen Ji In Il Wol Do) states that humans who respond to the sky above and the Earth below interact with heaven, the Earth and all directions.
Thank you
Point 4: please authors describe exclusion criteria clearly as these should be planned before the recruitment.
Response 4:
I revised it as you requested.
- Insertion completed (line 122-125).
- (line 122-125) The lack of a serious illness and the consent to participate in the research were inclusion criteria; exclusion criteria comprised physical problems related to spinal cord injury, paralysis, history of antidepressant use, history of psychiatric disorders, history of hormonal therapy (HT) use and symptoms for all items of the MRS.
Thank you
Point 5: Authors should change the word “Corona” with “Covid”
Discussion
Response 5:
I revised it as you requested.
- Insertion completed (line 107).
- (line 107) Corona --> Covid-
Thank you
Point 6: Please authors complete the reference n°48
Response 6:
I revised it as you requested.
- Insertion completed (line 261).
- (line 261) North American Menopause Society, 2014. --> The North American Menopause Society. Menopause21, 2014. 1063–1068.
Thank you
Reviewer 3 Report
Dear Jiyoun Kim,
This study is very interesting, I attach some small considerations for improvement.
Best regards.

Author Response
Response to Reviewer 3 Comments
Appreciate for your insightful and useful comments.
We’ve revised to reflect your comment.
Point 1: In the abstract, it would be interesting to specify the type of study and that it is randomized.
Response 1:
I revised it as you requested.
- Insertion completed (line 18-20).
- (line 18-20) Materials and methods: The participants comprised 40 women aged 50–65 years who were divided into two randomly selected groups in training sessions (exercising group, n = 21 and control group, n = 19).
Thank you
Point 2: “transition is a natural transition”… very repetitive transition, could be replaced by a synonym (process).
Response 2:
I revised it as you requested.
- Insertion completed (line 36).
- (line 36) menopausal transition is a natural transition --> menopausal transition is a natural change
Thank you
Point 3: “reducing”, “reduce” very repetitive, please use synonyms.
Response 3:
I revised it as you requested.
- Insertion completed (line 52).
- (line 52) reducing tension and anger to reduce depression, --> reducing tension and anger to decrease depression,
Thank you
Point 4: “The advantage of the program is the accessibility to the program”, very repetitive (program).
Response 4:
I revised it as you requested.
- Insertion completed (line 95).
- (line 95) The advantage of the program is the accessibility to the program, --> The advantage is the ease of accessing the program,
Thank you
Point 5: the final point is missing at the end of the paragraph.
Response 5:
I revised it as you requested.
- Insertion completed (line 114).
- (line 114) ~ limited spaces --> ~ limited spaces.
Thank you
Point 6: But how was the randomization done? It would be interesting to specify how the randomization was done.
Response 6:
I revised it as you requested.
- Insertion completed (line 117-119).
- (line 117-119) The present study was a randomized controlled trial. A total of 40 eligible women were recruited and randomly assigned to either a 12-week exercise intervention (three sessions/week) or to a usual care (control) group.
Thank you
Point 7: “The advantage of the program is the accessibility to the program”, very repetitive (program).
Response 7:
I revised it as you requested.
- We requested the MDPI editor to modify all texts.
Thank you
Point 8: In general, “this study” is very repetitive…perhaps it is better to write “in the present study or in the current study, to alternate on some occasions with “this study”.
Response 8:
I revised it as you requested.
- Insertion completed (line 2).
- (line 2) This study, --> In the current study,
- We requested the MDPI editor to modify all texts.
Thank you
Point 9: In general, “program” is very repetitive, could be replaced by a synonym.
Response 9:
I revised it as you requested.
- We requested the MDPI editor to modify all texts.
Thank you
Point 10: Author Contributions: at the end of the paragraph there is a point left over.
Response 10:
I revised it as you requested.
- Insertion completed (line 126).
- (line 116) K.M.H.,Y.E.S.; supervision, N.E.H; All authors approved the final version of the manuscript.. --> K.M.H.,Y.E.S.; supervision, N.E.H; All authors approved the final version of the manuscript.
Thank you
Round 2
Reviewer 2 Report
I think the authors have made good revisions to this paper. The paper's topic is worthwhile and interesting.
However references n° 6 and 7 are uncorrect (name instead of surname) and the revised references are:
6. De Franciscis, P.; Colacurci, N.; Riemma, G.; Conte, A.; Pittana, E.; Guida, M.; et al. A Nutraceutical Approach to Menopausal Complaints. Medicina (Kaunas). 2019, 55, 544, doi:10.3390/medicina55090544.
7. De Franciscis, P.; Conte, A.; Schiattarella, A.; Riemma, G.; Cobellis, L.; Colacurci, N. Non-hormonal treatments for menopausal symptoms and sleep disturbances: a comparison between purified pollen extracts and soy isoflavones. Curr. Pharm. Des. 2020, 26, doi:10.2174/1381612826666200721002022.
Author Response
Response to Reviewer 2 Comments
Appreciate for your insightful and useful comments.
We’ve revised to reflect your comment.
Point 1: Please remove the term middle-aged from the title and the text. The population can be referred to as menopausal women, and if you want to add further clarification, then the specific age range could be included.
Response 1:
As requested, the contents have been modified and supplemented.
- We have removed the term middle-aged woman from the title and the text.
Thank you
Point 2: references n° 6 and 7 are uncorrect (name instead of surname)
Response 2:
As requested, the contents have been modified and supplemented.
- (line 142-146) 6. De Franciscis, P.; Colacurci, N.; Riemma, G.; Conte, A.; Pittana, E.; Guida, M.; et al. A Nutraceutical Approach to Menopausal Complaints. Medicina(Kaunas). 2019 Sep; 55, 544.
- De Franciscis, P.; P.; Conte, A.; Schiattarella, A.; Riemma, G.; Cobellia, L.; Colacirci, N. Non-hormonal treatments for menopausal symptoms and sleep disturbances: a comparison between purified pollen extracts and soy isoflavones. Curr. Pharm. Des. 2020, 26.
Thank you